# Cytotaxonomy and Molecular Analyses of *Mycteria americana* (Ciconiidae: Ciconiiformes): Insights on Stork Phylogeny

**DOI:** 10.3390/genes14040816

**Published:** 2023-03-28

**Authors:** Rodrigo Petry Corrêa de Sousa, Paula Sabrina Bronze Campos, Michelly da Silva dos Santos, Patricia Caroline O’Brien, Malcolm Andrew Ferguson-Smith, Edivaldo Herculano Corrêa de Oliveira

**Affiliations:** 1Instituto de Estudos Costeiros, Universidade Federal do Pará, Bragança 68600-000, Brazil; rodrigo.correa@braganca.ufpa.br; 2Programa de Pós Graduação em Genética e Biologia Molecular, Universidade Federal do Pará, Belém 66075-110, Brazil; 3Cambridge Resource Centre for Comparative Genomics, Cambridge, CB3 0ES, UKmaf12@cam.ac.uk (M.A.F.-S.); 4Instituto de Ciências Exatas e Naturais, Universidade Federal do Pará, Belém 66075-110, Brazil; 5Instituto Evandro Chagas, Seção de Meio Ambiente, Ananindeua 67030-000, Brazil

**Keywords:** woody stork, chromosome painting, *cytochrome oxidase I*, *cytochrome b*

## Abstract

Although molecular information for the wood stork (*Mycteria americana*) has been well described, data concerning their karyotypical organization and phylogenetic relationships with other storks are still scarce. Thus, we aimed to analyze the chromosomal organization and diversification of *M. americana*, and provide evolutionary insights based on phylogenetic data of Ciconiidae. For this, we applied both classical and molecular cytogenetic techniques to define the pattern of distribution of heterochromatic blocks and their chromosomal homology with *Gallus gallus* (GGA). Maximum likelihood analyses and Bayesian inferences (680 bp *COI* and 1007 bp *Cytb* genes) were used to determine their phylogenetic relationship with other storks. The results confirmed 2n = 72, and the heterochromatin distribution pattern was restricted to centromeric regions of the chromosomes. FISH experiments identified fusion and fission events involving chromosomes homologous to GGA macrochromosome pairs, some of which were previously found in other species of Ciconiidae, possibly corresponding to synapomorphies for the group. Phylogenetic analyses resulted in a tree that recovered only Ciconinii as a monophyletic group, while Mycteriini and Leptoptlini tribes were configured as paraphyletic clades. In addition, the association between phylogenetic and cytogenetic data corroborates the hypothesis of a reduction in the diploid number throughout the evolution of Ciconiidae.

## 1. Introduction

Storks are included in the family Ciconiidae (Ciconiiformes, Birds), a group of birds widely distributed in tropical and subtropical regions around the world [1,2]. Although storks are morphologically well-defined, the phylogenetic position of Ciconiidae is still controversial, even in the most recent molecular analyses [3,4,5,6]. The monophyly of Ciconiidae was recently corroborated by a study performed to determine their phylogenetic position, based on the analysis of *CR1* retrotransposon insertion [6]. The results also suggested that this family was the first group to diverge in a clade including other Ciconiiformes/Pelecaniformes [6].

In general, living species of storks are included in six different genera (*Mycteria, Anastomus, Leptoptilos Ephippiorhynchus, Jabiru,* and *Ciconia*) [7,8], grouped into three tribes: Mycteriini (*Mycteria* and *Anastomus*), Ciconiini (*Ciconia*), and Leptoptilini (*Leptoptilos*, *Ephippiorhynchus,* and *Jabiru*) [9]. However, several analyses based on osteological, morphological, behavioral, molecular, and cytogenetic data have generated controversy about the genera belonging to each tribe [8,9,10,11,12,13,14]. Despite the importance of comparative cytogenetic analyses in providing insights into avian phylogenetic studies and genome evolution, karyotypical data of Ciconiidae are limited to classical cytogenetic studies, many of them solely confined to a diploid number and chromosome morphology; diploid numbers range from 2n = 52 in *Ciconia nigra* to 2n = 78 in *Leptoptilus javanicus* [15,16]. As the morphology of macrochromosomes was found to be similar in most species, it was suggested that karyotype evolution in Ciconiidae mainly involved fusions between microchromosomes [15,16].

The wood stork (*M. americana*) is the only representative of this genus found in the American continent (from North America to central Argentina) [17]. According to the IUCN, the current numbers of *M. americana* globally listed are not of concern. However, in regions such as Mexico and the U.S., the populations of this species have been reduced due to habitat loss. Thus, the wood stork is listed as a species in “special protection” in Mexico, while in the U.S. it is listed as threatened [17,18].

In order to understand the diversity and conservation of different populations of *M. americana*, the genetic variability of this species has been extensively studied by molecular biology [19,20,21]. However, the available cytogenetic data are restricted to the characterization of the karyotype by classical cytogenetics, with a 2n = 72 [22].

Comparative cytogenetics using chromosome painting has substantially increased knowledge about karyotype evolution and phylogenetic relationships in birds by revealing synapomorphic chromosomal rearrangements [23,24,25]. Up to now, only two species of Ciconiidae have been analyzed by comparative chromosome painting with probes of *G. gallus* and *Leucopternis albicollis*: *Jabiru mycteria* and *Ciconia maguari* [8]. Both species have shown a fusion between homologs of GGA8/GGA9, suggesting that this is a synapomorphy for this family.

It is apparent that it would be helpful to analyze more species in this group so that their phylogenetics can be more precisely defined. This study uses chromosome painting with whole chromosome probes of *G. gallus* in *M. americana*. Additionally, we explore available data on mitochondrial gene sequences (*cytochrome b—Cytb* and *cytochrome oxidase I—COI*) of species of storks and integrate the comparative cytogenetic results with molecular data to present a phylogenetic hypothesis that includes other members of Ciconiidae.

## 2. Materials and Methods

### 2.1. Samples, Cell Culture, and Chromosome Preparations

Experiments were approved by the ethics committee (CEUA, Instituto Evandro Chagas, 42/2019). Feather pulp samples were collected from a male of *M. americana* maintained at Parque Mangal das Garças (Belém, PA, Brazil) and transported in ice to the laboratory at Instituto Evandro Chagas. Cell culture was performed according to Sasaki et al. [26] with modifications. The material was extracted from the feather in a clean Petri dish and mechanically dissociated with the use of scalpel blades before incubation in collagenase solution (0.0465 g/mL DMEN) for approximately 1 h. Afterward, the material was centrifuged, and the supernatant was discharged and substituted by 5 mL of DMEM (Sigma-Aldrich, MO, USA) supplemented with 10 % bovine fetal serum (Gibco, Waltham, MA, USA) and 5% Aminiomax (Invitrogen, Carlsbad, CA, USA). Culture flasks were maintained at 37 °C in a 5% CO_2_ incubator. Chromosomes were obtained after exposition to colcemid (0.0016 %, 1 h), hypotonic treatment (0.075 M KCl, 40 min), and fixation and washes with Carnoy fixative (3 methanol:1acetic acid), following standard procedures. Chromosome suspensions were kept in a freezer at −20 °C.

### 2.2. Classical Cytogenetics Experiments

Diploid numbers and chromosome morphology were determined by the analysis of 30 Giemsa-stained (5 % solution in buffer pH 6.8 for 5 min) metaphase plates. Chromosome morphology followed Guerra [27]. C-banding following Sumner [28] was performed to analyze the distribution of constitutive heterochromatin blocks. Slides were analyzed and digitally captured using a Leica DM1000 microscope (100× objective) coupled to a computer with the GenAsis software, version 7.2.6.19509 (Applied Spectral Imaging, Carlsbad, CA, USA).

### 2.3. Fluorescent In Situ Hybridization (FISH) Experiments

We applied 11 chicken (*G. gallus*) whole chromosome probes, corresponding to autosome pairs 1 to 11. These probes were obtained by flow sorting at the Cambridge Resource Center for Comparative Genomics (Cambridge, UK), amplified by DOP-PCR, and labeled by biotin or fluorescein. Experimental conditions followed de Oliveira et al. [29]. Approximately 10 metaphases were analyzed and registered for each probe using a fluorescence microscope Zeiss Axio Imager 7.2 (Carl Zeiss, Jena, Germany) and the software Axiovision 4.8 (Zeiss, Jena, Germany).

### 2.4. Molecular Phylogenetic Analysis

Two mitochondrial gene sequences (fragment size of 680 bp for *COI* and 1007 bp for *Cytb*) were obtained from the GenBank and/or Boldsystems (Appendix A). The data corresponded to 17 species of storks, and we also used one species of Ardeidae and one species of Threskiornithidae as outgroups.

Sequence alignments were performed using the default settings of Clustal W [30] implemented in the software BioEdit Version 7.2 [31] and later manually edited. Maximum likelihood (ML) and Bayesian inference (BI) phylogenetic analyses were performed using the Iqtree [32] and MrBayes 3.2.7 [33] software, respectively. The evolutionary model for each gene was selected using the Bayesian information criterion in the jModelTest2 software [34]. For both analyses, the chosen model was the GTR+I+G.

Regarding the parameters of the analyses, the ML analysis and the confidence of the branches of the best tree were analyzed in detail based on an ultrafast analysis of 10,000 bootstrap pseudoreplicates. In turn, the IB was based on four Markov chain runs for 10,000,000 generations, with trees being saved every 10 generations and 10% of the first trees discarded as burnt. Run performance and effective sample sizes (ESS > 200) were shown in Tracer 1.7.1 [35]. Finally, the topologies generated by both analyses were visualized and edited in FigTree, version 1.4.4 [36].

## 3. Results

### 3.1. Karyotype Characterization

We found 2n = 72 in *M. americana*, with 12 macrochromosome pairs (11 autosome pairs and sex chromosomes ZZ) and 24 pairs of microchromosomes. Pairs 1, 2, and 5–7 were submetacentric, pairs 8 and 10 were metacentric, pairs 3 and 9 were telocentric, and pair 11 was acrocentric. Sex chromosome Z was submetacentric (Figure 1).

C-banding revealed blocks of constitutive heterochromatin distributed and restricted to the pericentromeric region of macrochromosomes and some microchromosome pairs (Figure 2).

### 3.2. Chromosome Painting

Probes corresponding to GGA1-11 produced 12 signals in the karyotype of *M. americana* (Figure 3). Probes GGA1, 2, 3, 5, and 7 hybridized on MAM 1, 2, 3, 5, and 9, respectively. GGA4 hybridized on two distinct pairs, MAM4 and MAM10, while GGA 6 produced signals in MAM6q. GGA8 and GGA9 were found fused, corresponding to MAM7q and MAM7p, respectively. GGA10 corresponded to a microchromosome pair (MAM11), while GGA11 hybridized on MAM8 and MAM10. None of the probes used in the experiments have produced signals in MAM6p nor MAM8p, suggesting the occurrence of fusions involving ancestral microchromosomes.

### 3.3. Phylogenetic Analysis

The data set included 65 sequences available at GenBank and/or Boldsystems, corresponding to fragments of *COI* and *Cytb* (Appendix A). Both phylogenetic analyses resulted in a tree with the same topology with high node values, supported by bootstrap and posterior probability (Figure 4).

Ciconiidae was recovered as a monophyletic group, with five strongly supported monophyletic clades corresponding to each genus of the family (*Anastomus, Mycteria, Leptoptilos, Ephippiorhynchus, Jabiru,* and *Ciconia*), as expected. The clade with *Anastomus* was the most basal one and sister group to the other ciconids. In turn, species of the genus *Leptoptilos* and *Mycteria* formed groups isolated from the other ciconids, while the clade with species of the genus *Jabiru* grouped together as a sister group to *Ephippiorhynchus*, which corresponded to a sister group of *Ciconia*, although not strongly supported.

The cytogenetic data, when associated with the respective species, showed similarity regarding the configuration of the diploid number for each group, where phylogenetically closer species presented similar or approximated diploid values (Figure 4). In addition, we observed a tendency of reduction in diploid number throughout ciconid diversification, with the most basal species having 2n ≥ 70.

## 4. Discussion

Although cytogenetic data concerning Ciconiidae are still scarce, it is clear that this family has interesting karyotypic features, given the great diversity in diploid numbers (2n = 52 to 2n = 78) and evidence of chromosomal rearrangements with potential phylogenetic value [8,37]. In this sense, considering that chromosomal rearrangements play a key role in speciation and genome evolution, whole chromosome probes provide important insights into the karyotypical diversification of birds [38,39,40].

Regarding previous karyotypic data, although the diploid number of 2n = 72 described by us for *M. americana* is identical to that found by Francisco and Galetti [22], our results differ in relation to the morphology of the macrochromosomes. This difference may be related to differences due to the degree of condensation of the chromosomes. In addition, the C-banding pattern of *M. americana* revealed a distribution of heterochromatic blocks only in the centromeric region, without any interstitial blocks. This pattern is similar to other groups of birds [39,41,42]. However, in *Ciconia ciconia*, the only species of Ciconiidae with a C-banding pattern, pairs 7, 8, and chromosome W seemed to be entirely heterochromatic [37].

Compared with the ancestral avian karyotype, it is possible to observe that the decrease in diploid numbers is related to a concomitant decrease in the number of microchromosomes, while macrochromosomes maintain similarity in number and morphology [8,22,37]. Although studies indicate that most bird species maintain evolutionary stability in their microchromosomal organization, in some orders (for example, Falconiformes, Psittaciformes, Cuculiformes, Trogoniformes, and Suliformes), fusions involving microchromosomes have been detected, resulting in small/medium biarmed chromosomes and consequent decrease in the diploid number [23,24,29,43]. However, although there is evidence of fusions involving microchromosomes in *M. americana* (GGA6/MIC and GGA11/MIC), the reduction in microchromosomes was not accompanied by an increase in the number of macrochromosomes, which leads us to conclude that the reduction in diploid numbers in some species of Ciconiidae is due to fusion events involving exclusively microchromosomes, both with other microchromosomes and with macrochromosomes [8].

Chromosome painting using *G. gallus* probes in *M. americana* showed results similar to what was observed in *J. mycteria* and *C. maguari*, suggesting the syntenies of macrochromosome pairs are conserved in Ciconiidae species [8]. In addition, GGA8/GGA9 and GGA6/microchromosome fusions were also found in *M. americana*, possibly corresponding to represent synapomorphies for the group, as postulated before [8] (Figure 5). Although there is still no evidence of shared fusions between Ciconiidae and other Ciconiformes families, GGA7/GGA8 fusion has been detected both in Treskiornithidae and Ardeidae, reinforcing the proposition that places herons and ibises in a single clade, as reported in molecular phylogenomic [25].

The integration of the cytogenetics data and molecular analyses based on *Cytb* and *COI* resulted in a coherent relationship between the species and reinforced the hypothesis of a tendency of reduction in the diploid number during the evolution of storks. Despite the fact of only using two genes, this is the first study investigating the molecular phylogeny of Ciconiidae. This issue has been a major problem, because most of the existing analyses were only based on morphological and behavioral data [10,11,12,13,14].

Our phylogenetic study recovered Ciconinii as a monophyletic group, while Mycteriini and Leptoptlini were configured as paraphyletic clades. In addition, although the clades formed by *Anastomus, Leptoptilos,* and *Mycteria* were placed as basal in the phylogeny of Ciconiidae, previous studies such as Slikas [11,12] and Pietri and Mayer [13], based on behavioral data and osteological and molecular measurements (*Cytb*), recovered only *Anastomus* and *Mycteria* as a basal and monophyletic group. Here, it is important to emphasize that there are few characters that justify the monophyly of Mycterinni. Hence, even though *Anastomus* shares some synapomorphies with other storks, this genus has a very distinct morphology, not allowing an obvious association with any genus [13,14]. In turn, as noted by Slikas [11,12], *Jabiru* and *Ephippiorhynchus* are phylogenetically closer than *Leptoptilos*. Thus, our data highlight the study by Selligman et al. [8], who based on cytogenetic data suggested the creation of a fourth tribe (the Ephippiorhynchini tribe, composed of *Ephippiorhynchus* + *Jabiru*), leaving the Leptoptlini tribe with only species of the *Leptoptilos* genus.

## 5. Conclusions

Our results indicate that chromosomal fusions involving microchromosomes played an important role in the karyotypic evolution of Ciconiidae species, in addition to macrochromosome fusions, which seem to represent chromosomal synapomorphies that may clarify the phylogenetic relationships of Ciconiidae. Using comparative chromosome painting and molecular analyses based on *Cytb* and *COI*, the phylogenetic proposal presented here shows a clear relationship between the species of Ciconiidae, which reinforced a tendency for the diploid number to decrease during the diversification of storks. However, the need for more chromosome painting data from other ciconids is evident, along with the availability of more molecular data to better understand and corroborate the evolutionary relationships within the group.

## Figures and Tables

**Figure 1 genes-14-00816-f001:**
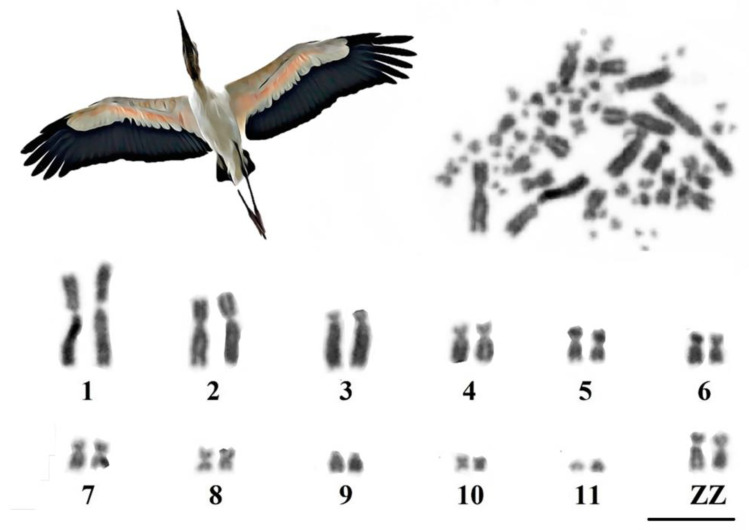
Photomicrograph of a spread of a male *M. americana* in conventional staining (top) and partial karyotype (bottom) with macrochromosomes arranged according to their size (bar = 10 µm).

**Figure 2 genes-14-00816-f002:**
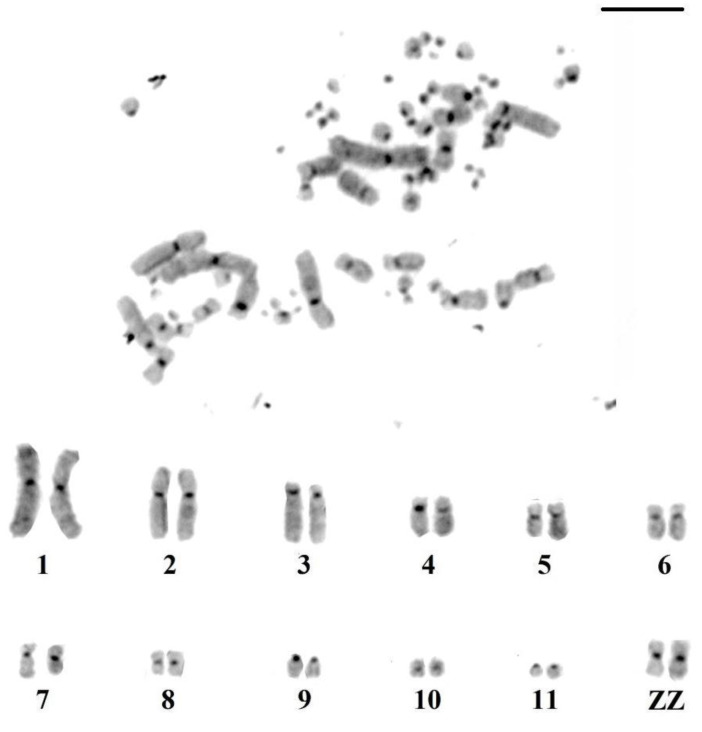
Distribution of constitutive heterochromatin in the karyotype of *M. americana* (bar = 10 µm).

**Figure 3 genes-14-00816-f003:**
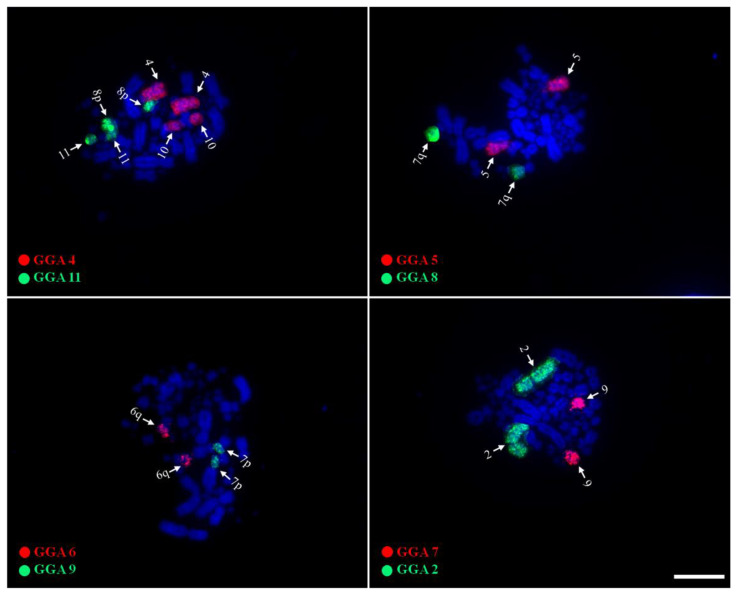
Representative FISH experiments using GGA whole chromosome probes on metaphase chromosomes of *M. americana* (bar = 10 µm). Arrows indicate precisely the chromosomes by each probe.

**Figure 4 genes-14-00816-f004:**
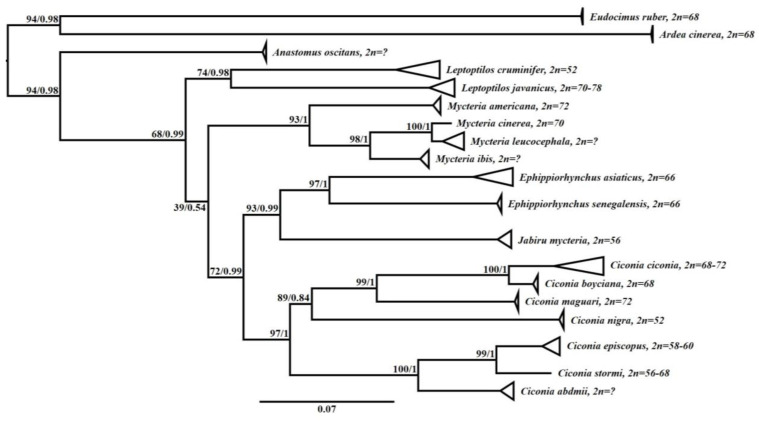
Phylogeny of Ciconiidae based on molecular and cytogenetic data. Node values correspond to support value (ML) and posterior probability (BI), respectively.

**Figure 5 genes-14-00816-f005:**
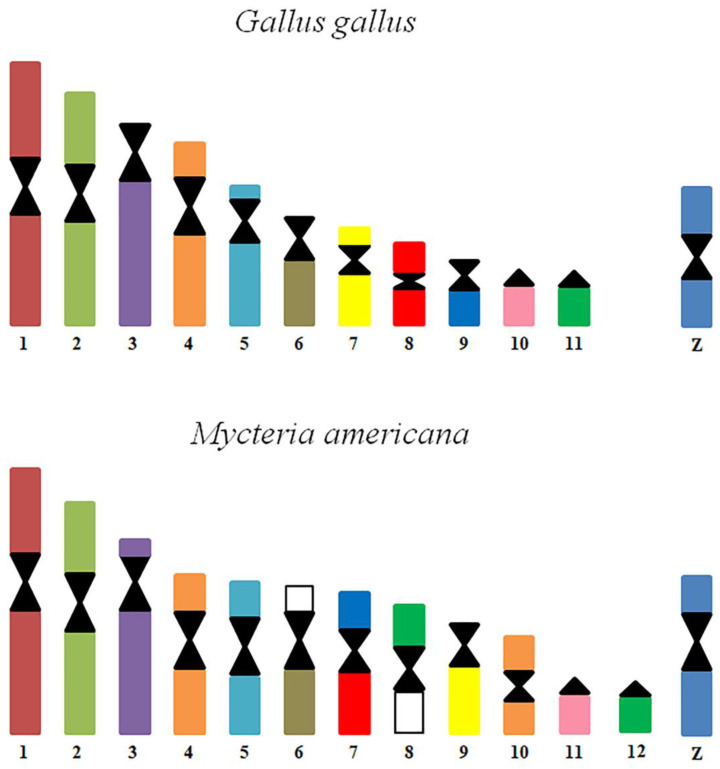
Homology map between *M. americana* and *G. Gallus*. The regions with homologies are indicated by color.

## Data Availability

All data from this study are available in the manuscript and Appendix A. In addition, other information can be obtained by request from the corresponding author.

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
