# Peer review of "Cytotaxonomy and Molecular Analyses of Mycteria americana (Ciconiidae: Ciconiiformes): Insights on Stork Phylogeny"

_genes, 2023, doi:10.3390/genes14040816_

Round 1
Reviewer 1 Report
Overall this is a correctly writen and executed manuscritp on the chromosomal organization and phylogenetic relationships of the wood stork, which provides valuable insights into the diversification of Mycteria americana. I found the combined use of "classical" and "modern" molecular cytogenetic techniques to be particularly informative, as it allows to identify fusion and fission events involving chromosomes homologous to GGA macrochromosome pairs. Coming from an absolutely different background and model animals I've always found particularly interesting the fact that michrochromosomes may rearrange into machrochromosomes so "easily".
Your phylogenetic analyses using COI and Cytb genes are correct, resulting in a tree that confirmed the monophyletic nature of Ciconinii, while Mycteriini and Leptoptlini tribes were configured as paraphyletic clades.
Overall, your research provides relevant evidence for the reduction in the diploid number throughout the evolution of Ciconiidae, and I commend you for your efforts.
Author Response
Dear Reviewer,
We, the authors, would like to thank you for your positive comments. Your comment is an incentive for us to seek to join our cytogenetic analysis with molecular data, with the aim of presenting more concise results which could contribute to the understanding of the phylogeny of Aves.
Reviewer 2 Report
Dear,
Thank you very much for choosing me as a review for the manuscript titled
” Cytotaxonomy and molecular analyses of Mycteria americana (Ciconiidae; Ciconiiformes): insights on the phylogeny of the storks’’
Manuscript ID: genes-2280980
This study aimed to analyze the chromosomal organization and diversification of M. americana, and provide evolutionary insights based on phylogenetic data of Ciconiidae.
The authors relied in the phylogeny of this study on samples from the GenBank and/or Boldsystems only and did not extract DNA or carried out PCR reaction even for one sample.
Also, the results of the sequences are not clearly stated in the heart of the research.
In abstract: the authors didn't mention the range of nucleotide length (bp.) or the
nucleotide averages.
In Materials and Methods: the authors did not explain in a table the scientific name, accession number, and reference for each sample from GenBank and/or Boldsystems.
In results, microchromosomes unmentioned in karyotype of M. americana (Figure 1). The microchromosomes should have been contained in the karyotype if possible.
*Resolution of Figure 1 is faint
*Morphometric analysis of chromosomes unmentioned
The authors didn't mention:
*The length of nucleotide sequence (bp.)
*The nucleotide frequencies (A, T, C and G) and their averages.
* Phylogeny based on molecular data should be separated from phylogeny based on cytogenetic data.
Again, the authors relied in the Phylogeny of this study on samples from the GenBank and/or Boldsystems only and did not extract DNA or carried out PCR reaction even for one sample.
Author Response
REVIEWER 2
Dear Researcher
The authors would like to thank you for your comments and suggestions, which will help us to improve the quality of our manuscript. We followed most of them and tried to justify the ones we did not follow. Here we present our answers to your questions.
Sincerely
The Authors
This study aimed to analyze the chromosomal organization and diversification of M. americana, and provide evolutionary insights based on phylogenetic data of Ciconiidae.
1.The authors relied in the phylogeny of this study on samples from the GenBank and/or Boldsystems only and did not extract DNA or carried out PCR reaction even for one sample.
R= GenBank and Barcode of Life Database (BOLD) are comprehensive databases that contains publicly available nucleotide sequences for more than 300 000 organisms named at the genus level or lower. For a broad spectrum of phylogenetic analyses, these online repositories proved to be important and reliable sources of sequence data. In addition, storks are distributed worldwide, and we had accessibility to only one species, Mycteria americana. We really do not see any reasons to perform the PCR of only this species if we would have to rely on the information taken from these resources concerning all the other species included in the molecular phylogeny
- Also, the results of the sequences are not clearly stated in the heart of the research.
R= The manuscript focuses mainly on the comparative chromosome paint, and the authors aimed to discuss the chromosome rearrangements that probably happened during the karyotypical diversification observed among storks. We added the molecular data as a plus, an accessory data that could guide our analysis, considering that most species do not have cytogenomic data, and the only information we have refers to their diploid number.
- In abstract: the authors didn't mention the range of nucleotide length (bp.) or the
nucleotide averages.
R= We added the information.
- In Materials and Methods: the authors did not explain in a tablethe scientific name, accession number, and reference for each sample from GenBank and/or Boldsystems.
R= These data are included in the supplementary material, which was uploaded in the submission process.
- In results, microchromosomes unmentioned in karyotype of M. americana (Figure 1). The microchromosomes should have been contained in the karyotype if possible.
R=, Usually, only macrochromosomes are ordered based on size and morphology. Because of their dot shape and small size, microchromosomes are left unordered in a whole metaphase figure.
6.*Resolution of Figure 1 is faint
R= We fixed it.
- *Morphometric analysis of chromosomes unmentioned
R= In Material and Methods, in line 99, you can find: . Chromosome morphology followed Guerra [27]
In Results (Lines 136-138), you can find: Pairs 1, 2, 5-7 were submetacentric, pairs 8 and 10 metacentric, pairs 3 and 9 telocentric, and pair 11 was acrocentric. Sex chromosome Z was submetacentric (Figure 1).
- The authors didn't mention:
8.aThe length of nucleotide sequence (bp.)
R= This information was found in the results (lines 165-166, The data set included sequences available at GenBank and/or Boldsystems, with a fragment size of 680 bp for COI and 1007 bp for Cytb), so we transferred it to the material and methods, and requested.
8.b*The nucleotide frequencies (A, T, C and G) and their averages.
R= We do not see any reason to mention this information considering these sequences were used solely to Maximum Likelihood (ML) and Bayesian Inference (BI) phylogenetic analyses
8c* Phylogeny based on molecular data should be separated from phylogeny based on cytogenetic data.
R= The molecular phylogeny was important to plot the cytotaxonomic data, in order to better understand the direction of the chromosome evolution (i.e. diploid number increase/decrease), considering that for most storks species we had only the diploid number.
- Again, the authors relied in the Phylogeny of this study on samples from the GenBank and/or Boldsystems only and did not extract DNA or carried out PCR reaction even for one sample.
R= We would like to reinforce that GenBank and/or Boldsystems proved to be important and reliable sources of sequence data. We really do not see any reasons to perform the PCR of only Mycteria americana if we would have to rely on the information taken from these resources concerning all the other species included in the molecular phylogeny.

Round 2
Reviewer 2 Report
The authors addressed all the comments.
All in all, the manuscript is well suited to be published in genes in current